# Class III Alcohol Dehydrogenase Plays a Key Role in the Onset of Alcohol-Related/-Associated Liver Disease as an S-Nitrosoglutathione Reductase in Mice

**DOI:** 10.3390/ijms241512102

**Published:** 2023-07-28

**Authors:** Takeshi Haseba, Motoyo Maruyama, Toshio Akimoto, Isao Yamamoto, Midori Katsuyama, Takahisa Okuda

**Affiliations:** 1Department of Legal Medicine, Kanagawa Dental University, 82 Inaokacho, Yokosuka 238-8580, Japan; yamamoto@kdu.ac.jp; 2Department of Legal Medicine, Nippon Medical School, 1-1-5 Sendagi, Bunkyo-ku, Tokyo 113-8602, Japan; 3Division of Laboratory Animal Science, Nippon Medical School, 1-1-5 Sendagi, Bunkyo-ku, Tokyo 113-8602, Japan; mmaru@nms.ac.jp (M.M.); toshio@nms.ac.jp (T.A.); 4Department of Legal Medicine, Kagoshima University Graduate School of Medicine and Dental Sciences, 8-35-1 Sakuragaoka, Kagoshima 890-8544, Japan; katsu@kufm.kagoshima-u.ac.jp; 5Department of Legal Medicine, Nihon University School of Medicine, 30-1 Oyaguchi-Kamicho, Itabashi-ku, Tokyo 173-8610, Japan; okuda.takahisa@nihon-u.ac.jp

**Keywords:** class III alcohol dehydrogenase, alcohol-induced hepatic steatosis, alcohol-related/-associated liver disease, chronic alcohol consumption, S-nitrosoglutathione reductase, peroxisome proliferator-activated receptor γ, mouse

## Abstract

Lipid accumulation in the liver due to chronic alcohol consumption (CAC) is crucial in the development of alcohol liver disease (ALD). It is promoted by the NADH/NAD ratio increase via alcohol dehydrogenase (ADH)-dependent alcohol metabolism and lipogenesis increase via peroxisome proliferator-activated receptor γ (PPARγ) in the liver. The transcriptional activity of PPARγ on lipogenic genes is inhibited by S-nitrosylation but activated by denitrosylation via S-nitrosoglutathione reductase (GSNOR), an enzyme identical to ADH3. Besides ADH1, ADH3 also participates in alcohol metabolism. Therefore, we investigated the specific contribution of ADH3 to ALD onset. ADH3-knockout (*Adh3-/-*) and wild-type (WT) mice were administered a 10% ethanol solution for 12 months. *Adh3-/-* exhibited no significant pathological changes in the liver, whereas WT exhibited marked hepatic lipid accumulation (*p* < 0.005) with increased serum transaminase levels. *Adh3-/-* exhibited no death during CAC, whereas WT exhibited a 40% death. Liver *ADH3* mRNA levels were elevated by CAC in WT (*p* < 0.01). The alcohol elimination rate measured after injecting 4 g/kg ethanol was not significantly different between two strains, although the rate was increased in both strains by CAC. Thus, ADH3 plays a key role in the ALD onset, likely by acting as GSNOR.

## 1. Introduction

Chronic alcohol consumption (CAC) causes alcohol-related/-associated liver disease (ALD) of varying severity, ranging from simple steatosis to more serious illnesses, such as steatohepatitis, fibrosis, cirrhosis, and hepatocellular carcinoma. Lipid accumulation in the liver is a crucial event in the onset of ALD [1,2]. However, the specific mechanisms involved in the development of ALD remain unknown. Lipids ectopically accumulated in the liver exhibit lipotoxicity, inducing insulin resistance and inflammatory cytokine release [3], and develop into ALD by co-acting with acetaldehyde and reactive oxygen (ROS) or nitrogen species (RNS) generated during alcohol metabolism [4,5,6].

CAC affects multiple lipid metabolism pathways in the liver by stimulating de novo lipogenesis, enhancing fatty acid uptake, suppressing fatty acid oxidation, and inhibiting very low-density lipoprotein export [7,8]. These effects of alcohol on hepatic lipid metabolism favor lipid accumulation in the liver.

Class I alcohol dehydrogenase (ADH 1), a key enzyme in alcohol metabolism, contributes to lipid accumulation in the liver by increasing the NADH/NAD ratio during alcohol metabolism, which further increases fatty acid levels in the liver by diverting acetyl-CoA toward fatty acid synthesis and inhibiting fatty acid β oxidation [4,9]. 

Peroxisome proliferator-activated receptor γ (PPARγ) is a master regulator of lipid metabolism via the expression of lipogenic genes, which is prominently distributed in adipose tissue [8]. PPARγ also contributes to lipid accumulation in fatty livers with or without alcohol by increasing liver content [3,8,10,11]. Zhang et al. [12] reported that PPAR-γ signaling develops ALD by promoting lipid accumulation and hepatic inflammation. Transcriptional activity of PPARγ on lipogenic genes is regulated by post-translational modification [11,13]. PPARγ is activated by acetylation and inactivated by the NAD-dependent deacetylase, sirtuin I (SIRT 1). However, CAC increases acetylated PPARγ levels in its active form as SIRT 1 activity is inhibited by an increase in the NADH/NAD ratio via ADH-dependent alcohol metabolism [11]. PPARγ is also non-enzymatically inactivated by S-nitrosylation with S-nitrosoglutathine (GSNO), a stable and mobile nitric oxide (NO), which is reduced to NH_3_ by S-nitrosoglutathione reductase (GSNOR). GSNOR is identical to the Class III alcohol dehydrogenase (ADH 3) [14], whose official gene symbol is *ADH* 5, as stated in the GenBank record (https://www.ncbi.nim.nih.gov/gene/11532 (accessed on 23 July 2023)). GSNOR also positively or negatively regulates various protein functions via denitrosylation of S-nitrosoproteins (protein-SNOs) [15,16]. Cao et al. [13] recently reported that PPARγ is activated by GSNOR in bone marrow-derived mesenchymal cells via denitrosylation and that it differentiates the cells into adipocytes by promoting adipogenesis. 

We recently reported that ADH 3, which possesses a very high Km for ethanol, also contributes to alcohol metabolism in a dose-dependent manner as an alcohol-metabolizing enzyme [17], as well as to CAC-induced alcohol metabolism with increasing liver content [18,19].

These multifunctions of ADH 3 are characteristic of its key role in lipid accumulation in the liver under CAC and in the development of ALD. However, there are no reports on the specific roles of ADH 3 or GSNOR in ALD, although a few studies have reported on their roles in non-alcoholic liver diseases [20,21,22]. Therefore, we investigated whether ADH 3 contributes to the onset of ALD in an animal CAC experiment using ADH 3-knockout (KO) (*Adh3-/-*) and wild-type (WT) mice for 12 months and explored the underlying mechanism.

## 2. Results

### 2.1. Base Line Characteristic of WT and Adh3-/- Mice during CAC (Body Weight, Survival Rate, Ethanol Intake and BAC)

Both strains of mice (WT and *Adh3-/-*) continuously increased their body weights in the control (C) and ethanol (E) groups during 12 months of CAC treatment. Body weights were significantly lower in *Adh3-/-* than in WT in both the C (7% lower, *p* < 0.001 by 2-way ANOVA) and E (11% lower, *p* < 0.0001) groups. When compared between C and E groups, the body weights of *Adh3-/-* were significantly lower in the E group than in the C group (5% lower, *p* < 0.001), whereas those of WT exhibited no significant difference between C and E groups (Figure 1a). *Adh3-/-* did not die during the CAC experiment, whereas WT started to die after four months of CAC and exhibited a 60% survival rate at the end of the CAC experiment (Figure 1b). The ethanol intake of *Adh3-/-* mice was almost constant during the CAC experiment, whereas that of WT mice gradually decreased and was 34% less than that of *Adh3-/-* mice after 12 months of treatment (Figure 1c). Blood alcohol concentration (BAC) measured near the end of the experiment was higher in *Adh3-/-* than in WT (*p* = 0.033 by Student’s *t*-test for *Adh3-/-* (5.92 ± 2.76) vs. WT (2.65 ± 0.71) at night), consistent with the ethanol intake results. The BAC was much higher during the day (*Adh3-/-*; 35.02 ± 7.76, WT; 27.62 ± 4.98) than at night in both strains (*p* < 0.001 for the day vs. night) (Figure 1d). 

### 2.2. ALD in WT But Not in Adh3-/- by CAC

The livers were removed from the mouse body, rinsed with saline, and evaluated for ALD. Comparing the gross images of the livers of WT and *Adh3-/-* mice after 12 months of the CAC experiment, WT livers were enlarged and whitish similar to a fatty liver, whereas *Adh3-/-* livers looked almost normal (Figure 2a). The ratio of liver weight to body weight in WT mice was significantly increased following CAC treatment (*p* < 0.001 for WT (E) (0.056 ± 0.007) vs. WT (C) (0.036 ± 0.005)). In contrast, that in *Adh3-/-* mice was not increased by CAC (Figure 2b), as it was already higher in the control group owing to the lower body weight (*Adh3-/-* (C) (0.055 ± 0.011), *Adh3-/-* (E) (0.059 ± 0.012)) (Figure 1a) [13,23]. Oil red O (ORO) staining showed marked lipid droplets in the liver tissue of WT (E) mice. However, the droplets were not noticeable in WT (C), *Adh3-/-* (E), and *Adh3-/-* (C) (Figure 3a). Even though the droplets were observed in the liver of these three groups, the sizes were smaller than those of WT (E). The rate of the ORO-positive area with lipid droplets >60 pixels was significantly higher in WT (E) (47.6 ± 11.2) than in WT (C) (16.1 ± 4.4), *Adh3-/-* (E) (8.0 ± 8.3), and *Adh3-/-* (C) (12.6 ± 3.3) (*p* = 0.0052 for WT (E) vs. WT (C), *p* = 0.0032 for WT (E) vs. the other groups), and was not different among WT (C), *Adh3-/-* (E), and *Adh3-/-* (C) (Figure 3b). A significantly higher rate of ORO-positive area in the WT (E) was observed at four months of CAC, compared with the other groups. Liver triglyceride (TG) levels after 12 months were significantly increased in WT (E), compared to WT (C) (*p* = 0.0027 for WT (E) (12.72 ± 2.46) vs. WT (C) (6.51 ± 1.75)). However, the levels were not significantly increased in *Adh3-/-* (E), compared to *Adh3-/-* (C) (*Adh3-/-* (E) (7.66 ± 1.79), *Adh3-/-* (C) (5.87 ± 1.55), *p* < 0.01 for WT (E) vs. *Adh3-/-* (C) or *Adh3-/-* (E)) (Figure 3c). Serum aspartate aminotransferase (AST) and alanine aminotransferase (ALT) levels were significantly increased in WT, and unaffected in *Adh3-/-* after 12 months of CAC (AST: *p* = 0.027 for WT (E) (170 ± 60) vs. WT (C) (68 ± 26), ALT: *p* < 0.01 for WT (E) (206 ± 128) vs. WT (C) (29 ± 14)) (Figure 4). 

### 2.3. Changes in Alcohol Elimination Rate (AER) and ADH 3 mRNA Levels in WT Liver during CAC

Figure 5 shows the changes in AER for both strains during the CAC experiment. AER was significantly enhanced by CAC in both strains during all experimental periods (*p* < 0.0001 by 2-way ANOVA for WT (E) vs. WT (C) and for *Adh3-/-* (E) vs. *Adh3-/-* (C)). However, the difference in the AERs between WT and *Adh3-/-* was not obvious in either the C or E group. Despite the induction of ALD after four months of CAC in the WT(E) mice, the AER was still higher than that of WT (C) mice. However, the AERs decreased with age in both groups E and C for both strains (Figure 5). 

Figure 6 depicts *ADH 3* mRNA levels and GSNOR activity in WT livers during CAC. *ADH 3* mRNA levels were also significantly increased in the WT liver following the CAC treatment (*p* < 0.01 by 2-way ANOVA for WT (E) vs. WT (C)) (Figure 6a). The GSNOR activity for GSNO as a substrate also showed an increasing tendency with CAC during the experiment (*p* = 0.019 by FTEST for WT (E) vs. WT (C)) (Figure 6b).

## 3. Discussion

Although ADH 3 has been reported to function as a GSNOR or glutathione-dependent formaldehyde dehydrogenase (GSH/FLDH) in various diseases [16,24,25], few studies have investigated its role in liver diseases. Cox et al. [21] reported that GSNOR exacerbated acetaminophen-induced liver disease by activating Keap 1 via denitrosylation and inhibiting the cytoprotective nuclear erythroid 2-related factor 2 (Nrf 2) pathways. We have also reported that ADH 3 exacerbates tetrachloride-induced liver fibrosis by increasing retinoic acid levels via retinol metabolism to activate hepatic stellate cells and suppress natural killer (NK) cells [22]. In contrast, we have reported that ADH 3 contributes to the protection of the liver from non-alcoholic steatohepatitis by maintaining cellular GSH levels as a GSH-generating/recycling enzyme in cooperation with Nrf 2 [20]. Therefore, the roles of ADH 3 in liver diseases vary according to liver condition, type of insult, abundance of reactive oxygen species, source and amount of NO production, and cellular redox status [6,26]. This inconsistent role of ADH 3 in liver disease may be due to its multiple functions. First, we investigated the role of ADH 3 in ALD via a CAC experiment using *Adh3*-/- mice, providing a 10% ethanol solution instead of water. The present CAC treatment led mice to consume <10 g ethanol/kg/day ad libitum together with lab chow containing fat calories amounting to 12.5% of the total calories [19], which was milder than that of the standard CAC method that uses a liquid diet (LD) containing 5% ethanol and forcibly administered ethanol to animals at 13–21 g/kg/day and fat calories amounting to 35% of the total calories [27]. Therefore, our CAC method took at least 4 months to induce ALD in WT mice and was able to continue the CAC experiment for mice for as long as 12 months, although the standard LD method usually takes <1 month to induce ALD and allows CAC experiments to last only a few months in rodents. WT mice began to die after four months and showed a 40% mortality rate at 12 months. Surviving WT mice exhibited ALD with marked lipid accumulation induced by CAC (Figure 2 and Figure 3). However, *Adh3-/-* mice did not exhibit ALD or other pathological conditions induced by CAC. These data indicate that ADH 3 plays a key role in the onset of alcohol-related fatty liver and development of ALD.

An increase in the NADH/NAD ratio due to ADH-dependent alcohol metabolism during CAC contributes to lipid accumulation in the liver by increasing fatty acids and inhibiting the activity of SIRT 1 (NAD-dependent deacetylase sirtuin 1) to increase acetylated PPARγ in its active form [11]. PPARγ induces hepatic steatosis through the expression of monoacylglycerol *O*-acyltransferase 1 gene (*MGAT 1*), a target gene of PPARγ for TG synthesis [11,28] (Figure 7), and induces inflammation in CAC [11]. ADH 1 is a key enzyme in alcohol metabolism [17] that increases the NADH/NAD ratio. However, this study demonstrated that the *Adh3-/-* mice resisted the induction of ALD by CAC despite possessing ADH 1, similar to the WT mice. Therefore, the onset of ALD may not be attributable to an increase in the NADH/NAD ratio due to ADH 1-dependent alcohol metabolism.

We previously reported that ADH 3 contributes to dose-dependent alcohol metabolism using originally generated *Adh3-/-* mice [17] and to CAC-induced alcohol metabolism by increasing its liver content [18,19]. However, the congenic *Adh3-/-* used in the present study, which was produced by backcrossing the original *Adh3-/-* mice with C57BL/6N mice of more than 13 generations, did not exhibit significant differences in AERs in WT during the CAC experiment in both the control and ethanol groups. This inconsistency in the AER between the original and congenic *Adh3-/-* mice may be explained by an increase in ADH 1-dependent alcohol metabolism in *Adh3-/-* mice during the production process of the congenic strain [19], probably compensating for the lack of ADH 3. Therefore, there may be no significant difference in the NADH/NAD ratio between the WT and *Adh3*-/- mice during CAC as there is no difference in AERs between the two strains, which mainly depends on ADH 1 and ADH 3 [19]. Therefore, an increase in the NADH/NAD ratio owing to ADH 3-dependent alcohol metabolism in the WT mice may not contribute to the total increase in the NADH/NAD ratio via ADH-dependent alcohol metabolism during CAC and to the ALD onset.

Besides the alcohol-metabolizing activity, ADH 3 also possesses GSNOR activity. GSNOR metabolizes GSNO so that S-nitrosylates various proteins to produce S-nitrosoproteins (protein-SNOs) nonenzymatically [25]. GSNOR also regulates the functions of S-nitrosoproteins via denitrosylation [15,16,21]. PPARγ, a key regulator of lipogenesis, is inhibited by S-nitrosylation with GSNO to form S-nitrosoPPARγ (PPARγ-SNO) but is activated by GSNOR via denitrosylation [13]. Deletion of *GSNOR* increases the levels of both GSNO and total S-nitrosoproteins in vivo [14]. *Adh3-/-* exhibits a lower body weight and fat levels than WT [13,19,23], probably because *Adh3-/-* cannot activate PPARγ via denitrosylation due to the lack of GSNOR; therefore, it cannot differentiate bone marrow-derived mesenchymal cells into adipocytes [13]. 

Although PPARγ generally distributes at low levels in the liver [29,30], it plays an important role in the induction of liver steatosis via the expression of *MGAT 1*, a key enzyme in TG synthesis, with increasing its liver content [11]. In this study, we demonstrated that only WT mice, but not *Adh3-/-*, exhibited ALD induced by CAC, with a significant increase in liver *ADH 3* mRNA and GSNOR activity during the experimental period. These data suggest that ADH 3 contributes to the onset of ALD via GSNOR activity, which further activates PPARγ via denitrosylation. Even though CAC increases acetylated PPARγ (Ac-PPARγ) levels in the liver by inhibiting SIRT 1 activity due to the increase in the NADH/NAD ratio, our results suggest that Ac-PPARγ-SNO is inactive in expressing *MGAT 1* without denitrosylation as the fatty liver was not induced in *Adh3-/-*, in which Ac-PPARγ-SNO cannot be denitrosylated because of the lack of GSNOR (Figure 7). These findings indicate that ADH 3 plays a key role in the onset of ALD, probably as a GSNOR, by activation of PPARγ in the liver via denitrosylation. We propose a new hypothesis regarding the cause of CAC-induced hepatic steatosis, as shown in Figure 7. In order to verify our hypothesis, further investigations are required to provide evidence that an inactive form of S-nitrosylated Pear is increased in *Adh3-/- mice*, thereby, depressing the expression of target genes, such as *MGAT 1*, in the liver during CAC, compared to the case of the WT mice.

Another mechanism may also be involved in the contribution of ADH 3 to the onset of ALD as GSNOR. GSNOR can lower the NO level in the liver sinusoidal microvessels, because ADH 3 abundantly locates in the endothelium of the liver sinusoid [31]. The resistance of *ADH3-/-* for ALD as shown in this study may be partly due to an increase in the level of NO in the liver, as Choi [16] reported that GSNOR-/- mice increase the NO level in the liver. It is well known that NO derived from sinusoidal endothelial cells is protective against the development of liver disease by maintaining the microcirculation of the liver [6,26,32] and that dysfunction of the microcirculation of the liver is one of the initiators of ALD [33]. Thus, ADH 3 also contributes to the induction of ALD as a GSNOR by metabolizing NO in the liver sinusoid. 

This study demonstrated that *Adh3-/-* mice exhibited no significant pathological changes in the liver after 12 months of CAC, whereas WT mice showed marked ALD with increasing liver ADH 3. Thus, it was revealed that ADH 3 contributes to the ALD onset. The contribution mechanism of ADH 3 to ALD may be attributed to its GSNOR activity, which activates PPARγ as a key regulator of ALD through denitrosylation and decreases NO level in the liver sinusoid.

In conclusion, ADH 3 plays a key role in the ALD onset, likely by acting as GSNOR, through possible mechanisms of the activation of PPARγ and the elimination of NO in the liver sinusoid.

## 4. Materials and Methods

### 4.1. Animals

*Adh3-* mouse was obtained from the Burnham Institute (USA) in 1999 [23]. A congenic strain of *Adh3-/-* was used, which was produced by backcrossing the original knockout mouse with a C57BL/6N strain as the background WT (Sankyo Lab., Co. Ltd., Tokyo, Japan) for up to 13 generations, and was maintained at Nippon Medical School in Japan. All animal experiments adhered to the ARRIVE guidelines and were approved by the Institutional Committee of Laboratory Animals of Nippon Medical School (approval No. 28-001) in accordance with the National Institutes of Health Guide for the Care and Use of Laboratory Animals (NIH Publication No. 8023, revised 1973) [19]. 

### 4.2. CAC Experiments

Male mice (C57BL/6N as WT, *Adh3-/-*) were selectively housed in cages (5–8 mice/cage) in a specific pathogen-free facility and provided with a standard mouse diet (MF pellets, Oriental Yeast Co., Ltd., Tokyo, Japan). The facility’s lights were turned on at 8 a.m. and turned off at 8 p.m. CAC experiments were performed on 9-week-old male mice by providing a 10% (*w*/*v*) ethanol/water solution ad libitum, instead of pure water, as previously described [19]. The CAC group (E group) was designated as WT (E) or *Adh3-/-* (E). The control group (C group) was provided with only water and designated as WT (C) or *Adh3-/-* (C). The experiment lasted for over 12 months and was divided into three periods. The number of mice in each group was 40 during the first period (the start of CAC to 1 month), 30 in the second period (from 1 month to 4 months), and 20 in the third period (from 4 months to 12 months), because 10 mice from each group were used for sample collection and measurements of AER at the end of each period. The number of WT (E) mice in the third period decreased from 20 to 12 owing to death, contrary to the other groups. Mouse weight, survival rate, daily alcohol intake, and BAC were measured as baseline characteristics during the CAC experiment [18,19]. The survival rate (%) of the mice was expressed as the ratio of the number of surviving mice to the initial number of mice of each group in each period. The measurement of alcohol intake of individual mouse is a study limitation in our CAC method, because 5–8 mice are housed together in a cage. Therefore, amount of alcohol intake of mouse during CAC experiment was estimated as follows. The reduced volume of alcohol solution was weakly measured in each cage. The total reduced volume of several cages of each group was divided by the total weights of the mice of each group and seven (days) to obtain the average amount of daily alcohol intake of mouse (g ethanol/kg of mouse body weight/day), which assumes that alcohol intake of a mouse is directly proportional its animal’s body mass. 

### 4.3. Pathological Analyses of Liver Tissue and Serum Samples

Mice were sacrificed by cervical dislocation and their livers were first fixed in 4% paraformaldehyde (1 day), then in 10% neutralized formalin, and embedded in an O.C.T compound (Tissue-Tek, Tokyo, Japan) for histopathological diagnosis. Frozen sections (5 µm) from lobus hepatis sinister lateralis were stained with ORO (Muto Pure Chemicals, Tokyo, Japan) and counterstained with HE. Using ImageJ image analysis software (V1.52), the rate of ORO-positive areas in the total liver cell area was measured using four photographs taken from a liver sample of each mouse under an objective lens with a magnification of 20×. The mean rate of the four photographs was designated as the ORO positivity rate. Lipid droplets <60 pixels were omitted from the calculation as they were also observed in the control groups. Pathological analyses of liver tissues were performed at New HistoScience Laboratory Co. Ltd. (Tokyo, Japan). Blood was collected from the vena cava, and the plasma levels of ALT and AST were measured using an Automatic Analyser 180 (Hitachi, Tokyo, Japan) at the Laboratory of Oriental Yeast, Co., Ltd. (Nagahama, Shiga, Japan).

### 4.4. Measurement of Liver TG Levels

Following Chamulitrat et al. [34], the livers obtained from mice after CAC were sonicated in 10 volumes of 2:1 chloroform/methanol, and lipids were extracted using Folch’s method. After centrifugation at 15,000 rpm for 10 min, the chloroform layers were collected and evaporated to complete dryness. Dried lipids were dissolved in 50 µL 3:2 hexane/isopropanol. TG levels (mg/g liver) were determined using a LabAssay^TM^ Triglyceride Kits (Wako), and a Multi-Microplate Reader (FilterMax F5; Molecular Devices, Sunnyvale, CA, USA).

### 4.5. Measurements of BAC and AER In Vivo

Blood (10 μL) was taken from the tail vein of the mouse around noon and 8 p.m. to measure BAC during the CAC experiment just before 12 months. It was immediately sealed in a vial containing 0.0025% n-propanol in saline solution as the internal standard, and the BAC was measured using gas chromatography (Parkin-Elmer Clarus580 GC) equipped with a head space sampler (Turb0Matrix HS40), as described previously [17,19]. To determine the AER, mice were injected with ethanol (26.8% *w*/*v* in physiological saline) at a dose of 4.0 g/kg between 9:00 and 11:00 a.m. the following morning after replacing 10% ethanol with water the night before. Blood (10 μL) was chronologically taken after ethanol injection. The mice were fed water only during blood sample collection. BAC was measured by GC, and AER (mg/kg/h) was evaluated by dividing the dose of ethanol (4.0 g/kg) by the time of alcohol metabolism (h), which was obtained from the point of intersection of the χ-axis and the regression line drawn by the linear least squares-method to fit to the pseudo-linear part (2–7 h) of the BAC-time curve, as previously described [17,19]. 

### 4.6. Quantification of ADH 3 mRNA Levels via Reverse Transcription-Quantitative Polymerase Chain Reaction (RT-qPCR)

*ADH 3* mRNA levels in the liver were determined by RT-qPCR analysis, as previously reported [18]. *Β-actin* mRNA was used as control. RNA sequences ACAGGAAAGAGTGCAGGATGG and TTGTGACCGGCAATCTCTCC (NM_007410.3, Mus musculus ADH 5 (class III), transcript variant 1) were used as the forward and reverse mRNA primers, respectively, to measure the *ADH 3* mRNA levels as ADH 3 is registered as *ADH 5* in GeneBank. RNA sequences of GCGCAAGTTAGGTTTTGTCAAAG and TGGATCAGCAAGCAGGAGTAC (NM_007393.5, Mus musculus actin, beta (Actb)) were used as the forward and reverse mRNA primers for the measurement of *β actin* mRNA.

### 4.7. Measurement of Liver GSNOR Activity

The GSNOR activity of ADH 3 was measured using GSNO as the substrate as previously reported [18,35]. Briefly, the liver extract was obtained by centrifugation at 15,000× *g* for 1 h at 2 °C after homogenization in six volumes of 50 mM Tris buffer (pH 8.0) containing 0.5 mM ethylenediamine tetraacetic acid. The liver supernatant was incubated at 30 °C with 75 μM NADH with or without 100 μM GSNO. NADH consumption was monitored using fluorescence spectrophotometry (FilterMax F%) with excitation at 340 nm and emission at 465 nm. The GSNOR activity was determined as the rate of NADH consumption in the presence of GSNO minus the rate of NADH consumption without GSNO.

### 4.8. Statistics

Data are presented as mean ± standard deviation (SD). Statistical significance was assessed using unpaired Student’s *t*-test (one-sided), two-way analysis of variance (ANOVA), or FTEST. Results were considered statistically significant at *p* < 0.05.

## Figures and Tables

**Figure 1 ijms-24-12102-f001:**
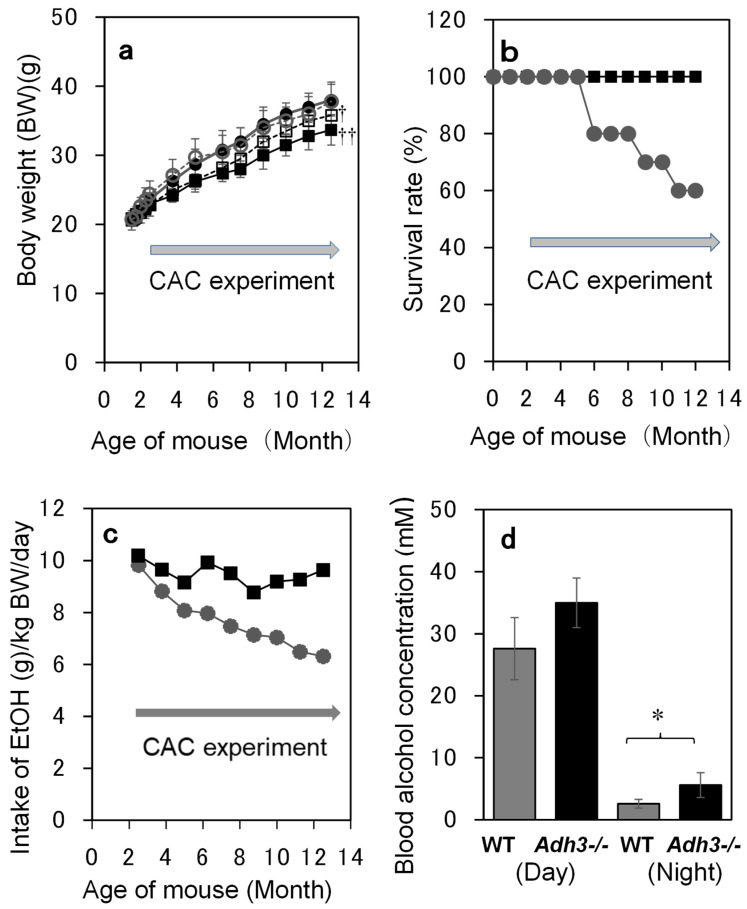
Baseline characteristics of WT and *Adh3-/-* mice during chronic alcohol consumption (CAC) experiment. Male mice were housed 5–8 each/cage. The ethanol groups {WT (E) and *Adh3-/-* (E)} were given 10% (*w*/*v*) ethanol solution ad libitum from 9 weeks old, whereas the control groups {WT (C) and *Adh3-/-* (C)} were given water, instead of ethanol. ●; WT (E), ○; WT (C), **■**; *Adh3-/-* (E), □; *Adh3-/-* (C). The number of mice in each group was 40 in the 1st period (from the start of CAC to 1 month); however, it decreased to 30 in the 2nd period (from 1 month to 4 months) and 20 in the 3rd period (from 4 months to 12 months), because 10 mice from each group were used at the end of each period for sample collections or measurements of alcohol elimination rate. (**a**) Body weights of WT and *Adh3-/-* mice. ^†^: *p* < 0.001 by 2-way ANOVA for *Adh3-/-* (C) vs. WT (C) and *Adh3-/-* (E) vs. *Adh3-/-* (C), ^††^: *p* < 0.0001 for *Adh3-/-* (E) vs. WT (E). n = 40 in the 1st period, n = 30 in the 2nd period, and n = 20 in the 3rd period. The number of WT (E) mice decreased from 20 to 12 due to death during the 3rd period, contrary to the other groups. (**b**) Survival rates of WT and *Adh3-/-* mice during CAC. Dots in the figure indicate the rate of surviving mice to the initial number of mice in each group at the beginning of each period. (**c**) Amount of alcohol intake of WT and *Adh3-/-* mice during CAC experiment. The reduced volume of alcohol solution in each cage was measured weekly. The total reduced volume of several cages in each group was divided by the total weights of the mice in each group and seven (days) to obtain the average amount of daily alcohol intake (g ethanol/kg of mouse body weight/day). (**d**) Blood alcohol concentration (BAC) in mice during the CAC experiment. BACs were measured at noon (day) and 8 p.m. (night), around 12 months of the experiment. *: *p* < 0.05 (n = 6~7).

**Figure 2 ijms-24-12102-f002:**
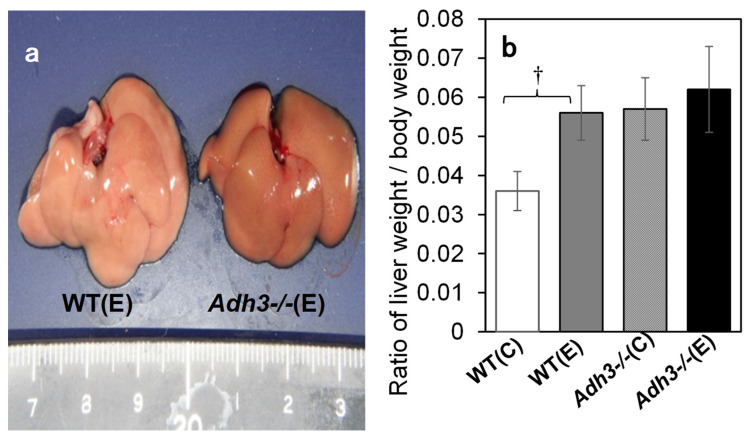
Liver steatosis of mice after 12 months of chronic alcohol consumption (CAC). (**a**) Gross picture of livers of WT and *Adh3-/-*. (**b**) Ratio of liver weight to body weight. (C) Water control; (E) 10% ethanol. n = 5. ^†^: *p* < 0.001.

**Figure 3 ijms-24-12102-f003:**
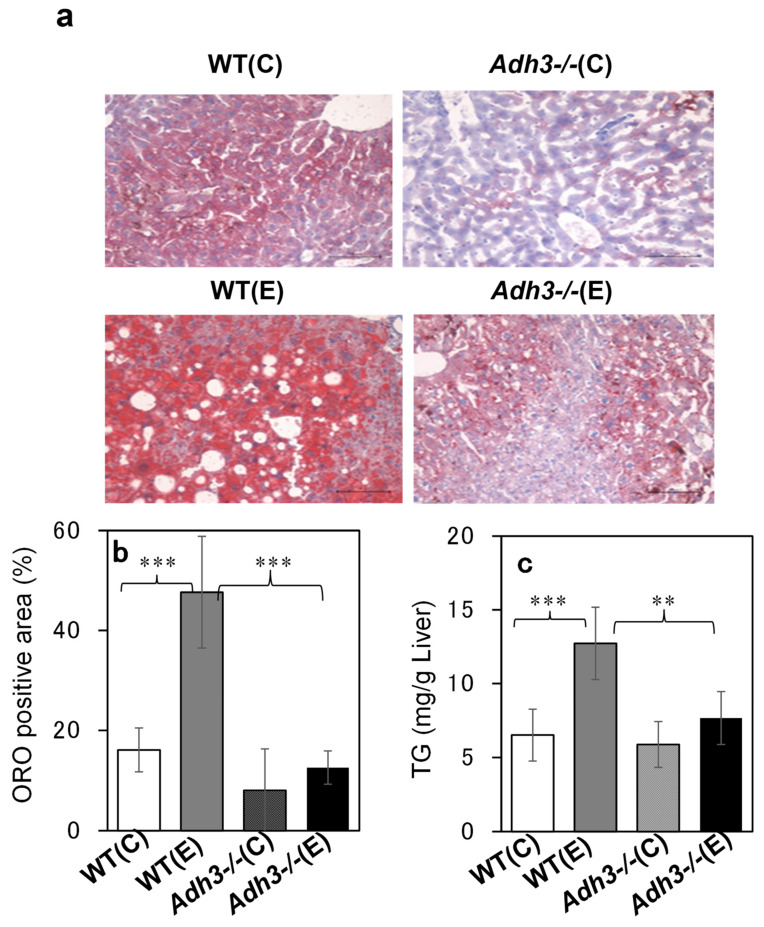
Oil red O (ORO) stains of liver tissues (**a**), the rate of ORO positive areas in liver cell areas (**b**) and liver contents of triglyceride (TG) (**c**) of mice after 12 months CAC. (**a**) Livers were fixed in 4% paraformaldehyde, followed by 10% neutralized formalin and embedded in O.C.T. compound (Tissue-Tek, Tokyo). Frozen sections of the lobus hepatis sinister lateralis were stained with ORO. (**b**) Lipid droplets smaller than 60 pixels in the image analysis software ImageJ (V1.52 )were omitted from the analysis, because they were also observed in the water control groups. The proportion of ORO-positive areas in the liver cell area was measured using four photographs of one liver sample from each mouse under an objective lens with a magnification of 20. The mean rate of the four photographs was designated as the ORO-positive rate. Three mice in each group were used for M ± SD (n = 3). See “Section 4 and the Data availability statement (**b**). (**c**) Lipids were extracted from the liver using Folch’s method, and TG levels in the livers were determined using LabAssayTM TG Kits (Wako, Japan). n = 5. **: *p* < 0.01, ***: *p* < 0.005, (C) water control, (E) 10% ethanol.

**Figure 4 ijms-24-12102-f004:**
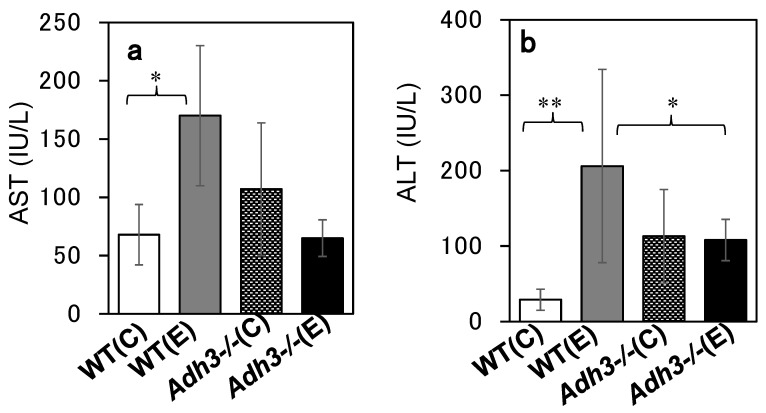
Serum AST (**a**) and ALT (**b**) of mice after 12 months of chronic alcohol consumption (CAC). (C) Water control, (E) 10% ethanol. *: *p* < 0.05, **: *p* < 0.01 (n = 5).

**Figure 5 ijms-24-12102-f005:**
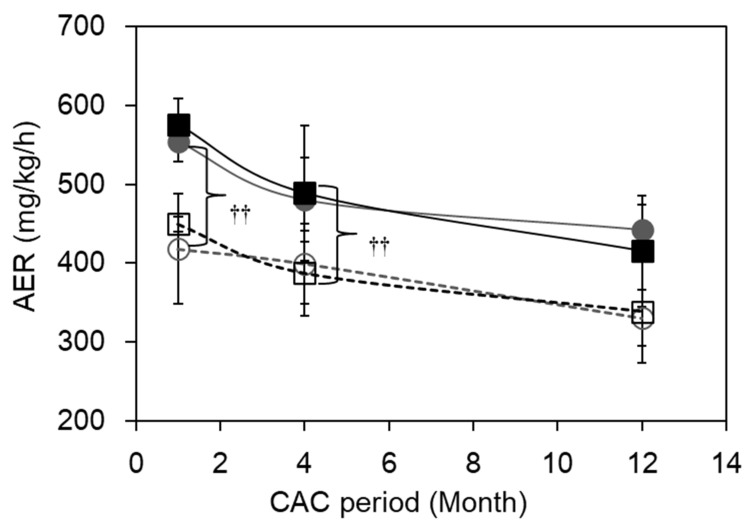
Alcohol elimination rate (AER) of mice during CAC. Mice were administered with ethanol at a dose of 4.0 g/kg. Blood ethanol concentration (BAC) was measured using blood sequentially collected from the tail. AER was calculated by dividing a dose of ethanol (4.0 g/kg) by hours of alcohol metabolism, which was obtained from the x-intercept of a regression line fitted to the pseudo-linear part of a curve of BAC by the linear least-squares method. ●: WT (E), ○: WT (C)**, ■**: *Adh3-/-* (E), □: *Adh3-/-* (C). **^††^**: *p* < 0.0001 by 2-way ANOVA. (n = 5 at each month).

**Figure 6 ijms-24-12102-f006:**
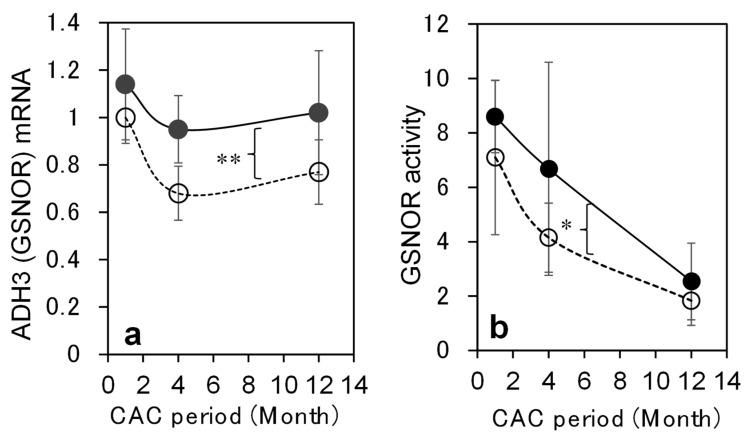
Changes of ADH3 (GSNOR) in the livers of WT mice during CAC. (**a**) ADH 3 (GSNOR) mRNA levels in WT livers during CAC. *ADH3* mRNA levels were expressed by the amount ratios of *ADH3* mRNA to *β-actin* mRNA. The amounts of *ADH3* and *β-actin* mRNAs were determined by a reverse transcription-quantitative PCR (RT-qPCR) analysis. (**b**) GSNOR activity in WT livers during CAC. Activity was measured using S-nitrosoglutathione (GSNO) as a substrate and expressed as unit/mg of liver protein. ●: WT(E), ○: WT(C). **: *p* < 0.01 by 2-way ANOVA, *: *p* < 0.05 by FTEST. (n = 5 per month).

**Figure 7 ijms-24-12102-f007:**
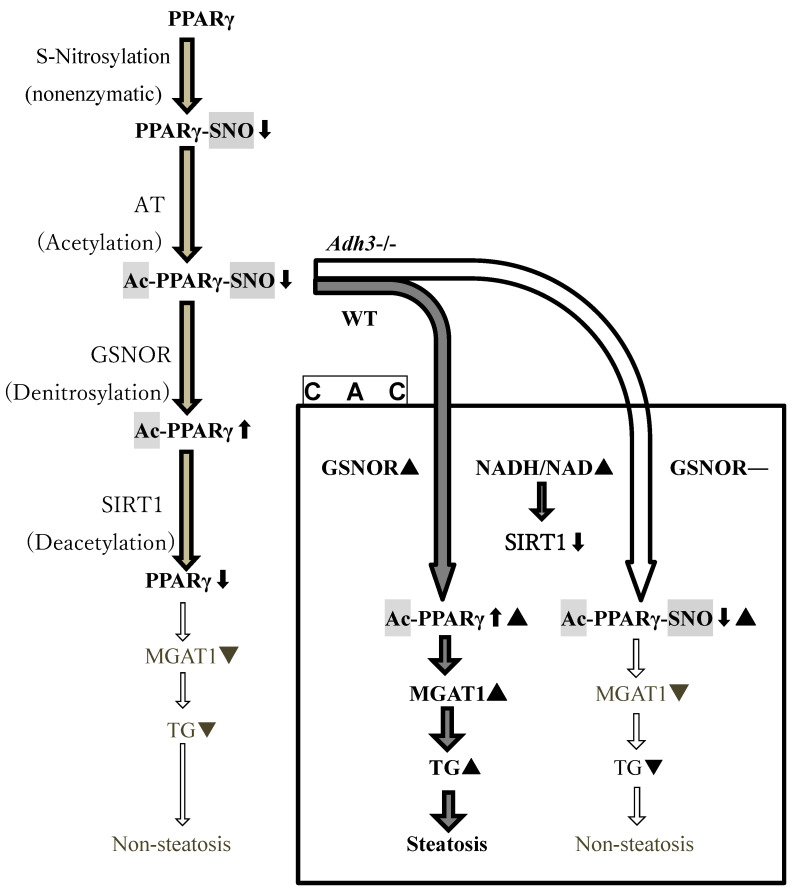
A hypothetical role of ADH 3 in the onset of alcohol-related/-associated liver disease as GSNOR. PPARγ, peroxisome proliferator-activated receptor; SNO, S-nitroso group; Ac, acetyl group; AT, acetyltransferase; *Adh3-/-*, ADH 3-KO mouse; WT, wild-type mouse; GSNOR, S-nitrosoglutathione reductase; CAC, chronic alcohol consumption; SIRT 1, NAD-dependent deacetylase sirtuin 1; MGAT 1, monoacylglycerol *O*-acyltransferase 1; TG, triglyceride; ▲, increase; **⬆**, active; ▼, decrease; **⬇**, inactive.

## Data Availability

Raw data of Figure 2, Figure 3 and Figure 4 and Figure 6 are available in Appendix A.

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
