# Peer review of "Class III Alcohol Dehydrogenase Plays a Key Role in the Onset of Alcohol-Related/-Associated Liver Disease as an S-Nitrosoglutathione Reductase in Mice"

_ijms, 2023, doi:10.3390/ijms241512102_

Round 1

Reviewer 1 Report

In this manuscript, Takeshi et al. conducted research to explore the role of GSNOR (also known as ADH5) in chronic alcohol consumption (CAC). Building upon previous work from the same group (PMID: 29663519), this study investigated the effects of global knockout of GSNOR in a more chronic model of CAC. The findings revealed that GSNOR knockout rescued liver injuries associated with CAC, adding to the existing knowledge on this topic. The manuscript is well-written, providing adequate introductions and discussions. However, it is worth noting that the amount of data presented may be insufficient to fully support the authors' hypothesis. There were no major issues identified in relation to the citations.

Here are some major issues:

1. The official gene symbol for GSNOR is Adh5, as stated in the GenBank record (https://www.ncbi.nlm.nih.gov/gene/11532). To avoid confusion, it would be beneficial for the authors to clarify this information in the introduction.

2. It is important for the authors to include a plot of GSNOR activity in the main figures (it is provide in the supplementary materials), as this data is crucial for supporting their conclusions.

3. While the manuscript presents changes in GSNOR mRNA and activities, it would be valuable to provide additional data demonstrating alterations in protein nitrosylation, particularly the nitrosylation of PPARg. This would strengthen the evidence supporting the proposed mechanisms.

4. The authors should include evidence of how PPARg activity is changing. This can be achieved by measuring the expression of PPARg target genes. Incorporating this data would provide a more comprehensive understanding of the impact of GSNOR on PPARg signaling.

5. To further investigate how lipogenesis is regulated in their model, the authors may consider measuring the expression of genes involved in this pathway, such as Fasn, Scd1, and others.

Here are some minor issues:

1. Page 4 The grammar is incorrect for the title “ALD in WT, but not in Adh3-/-, by CAC”

2. Figure 2 panel A and C, control liver needs to be shown for gross picture and H&E.

3. Figure 2, The size of picture needs to be adjusted to fit into the page

4. Oil Red O staining needs to be shown,

5. Figure 2 panel D, TG y-axis is blue color, “(” is missing

6. Figure 2 panel D, ORO positive area not matching H&E

7. Figure 4, y-axis title front is different

Additional recommendation:

Apart from lipogenesis, lipolysis in adipose tissue contributes to liver steatosis in alcoholic liver diseases. As GSNOR regulates adipogenesis (PMID: 25798618), is there any phenotype in adipose tissue in the GSNOR knockout mice?

Author Response

Author’s responses to the reviewers

Dear Sir (reviewer 1)

Thank you for reviewing our manuscript. I responded to your comments below.

  1. The official gene symbol for GSNOR is Adh5, as stated in the GenBank record (https://www.ncbi.nlm.nih.gov/gene/11532). To avoid confusion, it would be beneficial for the authors to clarify this information in the introduction.

I clarified the information in the introduction, as recommended.

  1. It is important for the authors to include a plot of GSNOR activity in the main figures (it is provided in the supplementary materials), as this data is crucial for supporting their conclusions.

A plot of GSNOR activity was shown in Figure 6 (b), together with GSNOR mRNA (a).

  1. While the manuscript presents changes in GSNOR mRNA and activities, it would be valuable to provide additional data demonstrating alterations in protein nitrosylation, particularly the nitrosylation of PPARg. This would strengthen the evidence supporting the proposed mechanisms.

I totally agreed to your comment. I understand that we need further investigations on PPARg activity and the levels of s-nitrosylation of PPARg in Adh3-/- mice to verify our hypothesis.

  1. The authors should include evidence of how PPARg activity is changing. This can be achieved by measuring the expression of PPARg target genes. Incorporating this data would provide a more comprehensive understanding of the impact of GSNOR on PPARg signaling.

 I agreed to you, as well as your comment of 3. we need to investigate on, such as, MGAT 1 expression levels in Adh3-/- mice liver in near future.

  1. To further investigate how lipogenesis is regulated in their model, the authors may consider measuring the expression of genes involved in this pathway, such as FasnScd1, and others.

As the same as your comment of 3 and 4, I agree to this comment. These molecular data on liver lipogenesis are also required to know how the PPARg-GSNOR system contributes lipogenesis in the liver, including NAFLD. I added a following sentence” further investigations are required to demonstrate that PPARγ is inactivated by denitrosylation not to express its target genes, such as MGAT 1, in the liver of Adh3-/- mice” in the Discussion.

Here are some minor issues:

  1. Page 4 The grammar is incorrect for the title “ALD in WT, but not in Adh3-/-, by CAC

I corrected the sentence to “ALD in WT(E) but not in Adh3-/- by CAC “.

  1. Figure 2 panel A and C, control liver needs to be shown for gross picture and H&E.

The pictures in the previous Fig. 2c were incorrectly labeled as H&E staining, although they are actually ORO staining pictures. This was my mistake. ORO pictures of four groups {WT(C), WT(E), Adh3-/-(C), Adh3-/-(E)} were presented in the new Fig. 3a. Gross pictures of WT(C) and Adh3-/-(C) livers were unfortunately not available, however, these were almost the same to the gross image of Adh3-/-(E), which was presented in the new Fig. 2a, together with that of WT(E).

  1. Figure 2, The size of picture needs to be adjusted to fit into the page

These pictures were properly presented in the new Fig. 3a as ORO staining pictures, and adjusted to fit into the page.

  1. Oil Red O staining needs to be shown,

ORO staining pictures of 4 groups were presented in the new Fig. 3a.

  1. Figure 2 panel D, TG y-axis is blue color, “(” is missing

The color was corrected to black and “(” was added in the new Fig. 3c.

  1. Figure 2 panel D, ORO positive area not matching H&E

One set of ORO staining of four mice groups was shown in the new Fig.3a. Four pictures were taken from one liver preparation of each group. Three preparations of the liver of each group were made from three mice of each group (n = 3).

  1. Figure 4, y-axis title front is different

The font of y-axis title in the previous Fig. 4a was properly corrected and presented independently in the new Fig. 5.

Additional recommendation:

    Apart from lipogenesis, lipolysis in adipose tissue contributes to liver steatosis      in alcoholic liver diseases. As GSNOR regulates adipogenesis (PMID:                25798618), is there any phenotype in adipose tissue in the GSNOR knockout mice?

 GSNOR-dependent modification of PPARγ alters the balance between adipocytes and osteoblasts differentiation. GSNOR KO mice exhibits decreased adipogenesis and increased osteoblastogenesis, consequently smaller weight with reduced fat mass and increased bone-formation. S-Nitrosylation of PPARγ was elevated in bone marrow-derived stem cells (MSCs) from GSNOR-/- mice, diminishing binding to its downstream target fatty acid-binding protein 4 (FABP4). (Ref. 12)   

Reviewer 2 Report

Following are my comments for the manuscript:

1) Too many grammar and language errors along with sentence formation throughout the abstract; needs to be re-written; it is difficult to pinpoint one error

2) Introduction is little waivered; language errors in few lines especially second last paragraph

3) Serious errors in figure 1 presentation; no significance is listed properly; the group description is at the very end of figure legend which is very confusing; figure 1b and c is missing significance values

4) Description in section 2.2 is confusing; what part of liver was used for H&E and ORO staining?; labelling of figure 2C is very poor; parenthesis is missing in Y-axis label of figure 2e. very poor presentation

5) Please explain big error bars in WT(C) vs WT(E) in figure 3a and 3b

6) Description of figure 4 has serious language errors; mRNA levels in figure 4 is important for Adh3-/- group as well to check the editing efficiency 

7) Discussion and abstract is heavily reliant on mentions of NADH/NAD ratios as well as PPAR gamma level changes; no experiments were performed regarding the same 

8) Most part of discussion is based on hypothesis and not dependent on conclusion of own work. Many experiments are required to support the hypothesis and connect it to the author's work

Please see comments from above section regarding the language error

Author Response

Dear Sir (reviewer 2)

Thank you for reviewing our manuscript. I responded to your comments below.

  • Too many grammar and language errors along with sentence formation throughout the abstract; needs to be re-written; it is difficult to pinpoint one error

English of our manuscript was recorrected by an English correction company “Editage for English language editing” (www.editage.com), based on the referee’s comment.

  • Introduction is little waivered; language errors in few lines especially second last paragraph

English in the introduction was also corrected as mentioned in 1).

  • Serious errors in figure 1 presentation; no significance is listed properly; the group description is at the very end of figure legend which is very confusing; figure 1b and c is missing significance values

The figure legend was re-written more concretely. The CAC experiment was divided into 3 periods, as written in the Materials & Methods in the revised manuscript. The mouse number of each group was 40 at the first period (the start of CAC to 1 month), decreased to 30 at second period (from 1 to 4 months) and to 20 at third period (from 4 to 12 months), because 10 mice of each group were used for sample collection and measurements of AER at the end of each period. Fig. 1b and c have no significance values, because the dots in Fig.1b indicates the rate of surviving mice to initial number of mice at the start of each period. In Fig 1c. the reduced volume of alcohol solution in each cage was measured every week. The total reduced volume of several cages in each group was divided by the total weights of mice in each group and seven (days) to obtain the average amount of alcohol intake (g ethanol/kg of mouse body weight/day). The number of mice of WT (E) at the 3rd period was decreased from 20 to 12 due to death, different from other groups.

  • Description in section 2.2 is confusing; what part of liver was used for H&E and ORO staining; labelling of figure 2C is very poor; parenthesis is missing in Y-axis label of figure 2e. very poor presentation

The description in section 2.2 was re-written. The lobus heptis sinister laterlis of the mouse liver was used for H&S and ORO staining, as described in the Materials and Methods (4.3).

Fig. 2c had been incorrectly labeled as H&E staining by my mistake, which is actually ORO staining. The Fig. 2c was transferred to the new Fig. 3a of ORO staining in the revised manuscript.

The parenthesis missing of Y-axis in Fig. 2e was corrected in the new Fig. 3c in the revised manuscript.

  • Please explain big error bars in WT(C) vs WT(E) in figure 3a and 3b

Indeed, WT (E) showed larger SDs of AST and ALT. As shown in the raw data of in Data availability Statement (Fig 4), each of SD contained an exceptional small value, respectively.

  • Description of figure 4 has serious language errors; mRNA levels in figure 4 is important for Adh3-/- group as well to check the editing efficiency 

Fig. 4a was independently transferred to the new Fig.5 and Fig.4b was transferred to the new Fig. 6 in the revised manuscript. The language errors were corrected by native. mRNA levels of ADH3 in Adh3-/- liver was not measured as expected no transcription, but the ADH3 protein level was previously revealed almost zero (Ref. 18).

  • Discussion and abstract are heavily reliant on mentions of NADH/NAD ratios as well as PPAR gamma level changes; no experiments were performed regarding the same 

PPAR gamma activity is inhibited by an increase in the NADH/NAD depending on alcohol metabolism (Ref. 10).

Indeed, we don’t have data on the inhibition of PPAR gamma activity by S-nitrosylation and MGAT1 expression in Adh3-/- liver. Our hypothesis was mainly based on the experimental evidence that Adh3-/- mice protects against ALD.

I understand that we need more molecular evidence to discuss the molecular mechanism of ALD involving PPAR gamma and GSNOR. We would try to response to your criticism by further investigations.

  • Most part of discussion is based on hypothesis and not dependent on conclusion of own work. Many experiments are required to support the hypothesis and connect it to the author's work

We present our hypothesis based on our finding that Adh3-/- mice resists ALD and on the previous data that GSNOR regulate PPARgamma.

As mentioned in 7), we need further investigations on behaviors of PPAR gamma-GSNOR system in the liver lipogenesis to verify our hypothesis. I added a following sentence” further investigations are requred to demonstrate that PPARγ is inactivated by S-denitrosylation not to express its target genes, such as MGAT 1, in the liver of Adh3-/- mice” in the Discussion.

Round 2

Reviewer 1 Report

The manuscript has shown significant improvement in terms of writing. However, one major concern is that the authors have not provided additional evidence to explain how ADH3 regulates PPARγ and steatosis, despite extensively discussing this in the introduction and discussion sections.

Furthermore, the authors have changed their conclusion to state that " ADH 3 plays a key role in ALD onset, like by acting as GSNOR." However, there is no supporting data to establish the significance of GSNOR activity in ALD.

(1) The authors mentioned that “GSNOR can lower NO level in the liver sinusoidal microvessels”. This commitment needs to be supported with evidence.

(2) The authors mentioned that “The resistance of ADH3-/- for ALD as shown in this study may be partly due to an increase in the level of NO in the liver, as Choi [15] reported that GSNOR-/- mice increase the NO level in the liver. It is well known that NO derived from sinusoidal endothelial cells is protective against the development of liver disease by maintaining the microcirculation of the liver [5, 27, 33]”. However, it should be noted that NO is also a reactive species that can induce liver damage and inflammation (see reference [5] in the manuscript).

Overall, the current manuscript does not provide sufficient evidence to convincingly support the role of ADH3 in ALD.

Author Response

Dear sir (Reviewer 1)

Thank you for reviewing our manuscript.

I tried to respond to your comments and criticisms below, as possible as I can.

-----------------------------------------------------------------------------------------

The manuscript has shown significant improvement in terms of writing. However, one major concern is that the authors have not provided additional evidence to explain how ADH3 regulates PPARγ and steatosis, despite extensively discussing this in the introduction and discussion sections.

I understand that our hypothetical role of GSNOR-PPARγ-MGAT1 system in ALD onset is needed to be supported by additional data, such as an increase in S-nitrosylated PPARγ and a decrease of MGAT1 expression in Adh3-/- mice during CAC, as pointed by you.

Nevertheless, we try to support this hypothesis here by our present data and other data previously published, as my present situation does not allow to produce additional data by performing further animal experiments.  

We provided evidence that ADH3 contributed to ALD onset, because ALD was induced in WT mice by CAC but not in Adh3-/- mice. The contribution mechanism of ADH3 to ALD onset was firstly assumed an increase in the NADH/NAD ratio by ADH3-dependent alcohol metabolism during CAC in WT (Ref. 3, 8). However, this assumption was negative, because AERs were not significantly different between WT and Adh3-/- during CAC (fig. 5). On the other hand, GSNOR is the same enzyme as ADH3 (Ref. 13) and regulates the activities of various enzymes by denitrosylation (Ref. 14, 15). PPARγ is also activated by denitrosylation via GSNOR and contributes to adipogenesis in the adipocytes as a key regulator (Ref. 12). PPARγ signaling pathway also play a key role in lipogenesis in the liver and in the onset of ALD by CAC (Ref. 10, 31), inducing steatosis and inflammation by increasing its liver content (Ref. 2, 11). The expression of MGAT1 by PPARγ is important for liver steatosis and ALD by CAC (Ref. 10, 11), because that AFL and ALD were not induced in PPARγ- knockdown mice (Ref. 11) and in MGTA1- knockdown mice (Ref. 10). Moreover, we showed in this study that WT mice developed ALD with increasing liver GSNOR by CAC, whereas Adh3-/- mice resists ALD onset. From these data, we considered that Adh3-/- mice did not induce ALD by CAC, because the mice cannot activate PPARγ for lack of GSNOR. The contribution mechanism of ADH3 to the ALD onset was illustrated in Fig. 7 as a hypothesis.

Furthermore, the authors have changed their conclusion to state that " ADH 3 plays a key role in ALD onset, like by acting as GSNOR." However, there is no supporting data to establish the significance of GSNOR activity in ALD.

(1) The authors mentioned that “GSNOR can lower NO level in the liver sinusoidal microvessels”. This commitment needs to be supported with evidence.

(2) The authors mentioned that “The resistance of ADH3-/- for ALD as shown in this study may be partly due to an increase in the level of NO in the liver, as Choi [15] reported that GSNOR-/- mice increase the NO level in the liver. It is well known that NO derived from sinusoidal endothelial cells is protective against the development of liver disease by maintaining the microcirculation of the liver [5, 27, 33]”. However, it should be noted that NO is also a reactive species that can induce liver damage and inflammation (see reference [5] in the manuscript).

I also understand that NO has two-edged sword in the liver. NO constantly produced from endothelial nitric oxide synthase (eNOS) in the liver sinusoid is protective, whereas the high levels of NO from inducible NOS (iNOS) in the parenchymal liver cells is harmful (Ref. 5, 27, Chen et al., 2003 Curr Mol Med).

A decrease of NO from sinusoidal non-parenchymal cells (NPCs) is important for the onset of ALD (Nanji et al.,1995 Gastroenterology). Alcohol intake, both acute and chronic, affects iNOS activity in various cells and inhibits inducible NO production (Ref. 27). GSNOR inhibitor and GSNOR-/- are hepatoprotective from acetaminophen liver injury and S-nitrosothiol signaling in sinusoidal NPCs is protective against liver cell injury (Ref. 20).

In our present study, ADH3-/- mice did not appear ALD by CAC. This result may partly owe to an increase in NO in the liver for lack of GSNOR (Ref. 15). The increase of NO in Adh3-/- liver may be not so high as harmful levels under our CAC condition, probably because NO comes from eNOS not from iNOS. From these data, we presented another possible mechanism of ADH3 as GSNOR for ALD onset, that is, GSNOR contributes to ALD onset by eliminating NO from eNOS in liver sinusoid, because ADH3 (GSNOR) is abundantly localized in sinusoidal endothelial cells (Ref. 32).

Overall, the current manuscript does not provide sufficient evidence to convincingly support the role of ADH3 in ALD.

We concluded from restricted data obtained in my present situation that ADH3 play a key role in ALD onset, like by acting as GSNOR, based on two possible mechanisms of ADH3, the activation of PPARγ and the elimination of NO from eNOS in the liver.

Reviewer 2 Report

The authors have heavily worked on the comments that I made previously to improve the manuscript presentation. Impressive work!

Author Response

Dear Sir (Reviewer 2)

Thank you very much for reviewing our munuscript.

Takeshi Haseba Ph.D.

Round 3

Reviewer 1 Report

Overall, small errors were corrected. Although the data is still limited, authors claim they will explore the remaining questions in the future. 

Author Response

Thank you very much for reviewing our manuscript repeatedly.